# Inhibition of Melanoma Cell Migration and Invasion Targeting the Hypoxic Tumor Associated CAXII

**DOI:** 10.3390/cancers12103018

**Published:** 2020-10-17

**Authors:** Gaia Giuntini, Sara Monaci, Ylenia Cau, Mattia Mori, Antonella Naldini, Fabio Carraro

**Affiliations:** 1Department of Molecular and Developmental Medicine, Cellular and Molecular Physiology Unit, University of Siena, 53100 Siena, Italy; gaia.giuntini@student.unisi.it (G.G.); sara.monaci@student.unisi.it (S.M.); Antonella.Naldini@unisi.it (A.N.); 2Department of Biotechnology, Chemistry and Pharmacy, University of Siena, 53100 Siena, Italy; Cau.Ylenia@gmail.com (Y.C.); mattia.mori@unisi.it (M.M.); 3Department of Medical Biotechnologies, Cellular and Molecular Physiology Unit, University of Siena, 53100 Siena, Italy

**Keywords:** carbonic anhydrase, hedgehog, cyclopamine, small molecules, acetazolamide, motility, metalloproteinases, FAK, cancer

## Abstract

**Simple Summary:**

Melanoma is a potential metastatic cancer with a poor prognosis and a low free-survival rate. Thus, discovering new therapeutic strategies will be helpful to fight against it. Carbonic anhydrases IX (CAIX) and XII (CAXII) along with Hedgehog pathway aberrations are already demonstrated to be involved in melanoma progression. Here we investigated the effects of either an indirect or direct CAXII inhibition on cell migration and invasion in three melanoma cell lines. First we indirectly inhibited CAXII through the smoothened antagonist cyclopamine, which resulted in a decreased CAXII protein expression and cell migration. Thereafter, we directly blocked CAXII using new small molecules. This resulted in reducing not only cell migration, but also the invasiveness ability of highly aggressive melanoma cell lines. This evidence may contribute to further exploiting the therapeutic role of CAXII in melanoma progression and invasiveness.

**Abstract:**

Background: Intratumoral hypoxia contributes to cancer progression and poor prognosis. Carbonic anhydrases IX (CAIX) and XII (CAXII) play pivotal roles in tumor cell adaptation and survival, as aberrant Hedgehog (Hh) pathway does. In malignant melanoma both features have been investigated for years, but they have not been correlated before and/or identified as a potential pharmacological target. Here, for the first time, we demonstrated that malignant melanoma cell motility was impaired by targeting CAXII via either CAs inhibitors or through the inhibition of the Hh pathway. Methods: We tested cell motility in three melanoma cell lines (WM-35, SK-MEL28, and A375), with different invasiveness capabilities. To this end we performed a scratch assay in the presence of the smoothened (SMO) antagonist cyclopamine (cyclo) or CAs inhibitors under normoxia or hypoxia. Then, we analyzed the invasiveness potential in the cell lines which were more affected by cyclo and CAs inhibitors (SK-MEL28 and A375). Western blot was employed to assess the expression of the hypoxia inducible factor 1α, CAXII, and FAK phosphorylation. Immunofluorescence staining was performed to verify the blockade of CAXII expression. Results: Hh inhibition reduced melanoma cell migration and CAXII expression under both normoxic and hypoxic conditions. Interestingly, basal CAXII expression was higher in the two more aggressive melanoma cell lines. Finally, a direct CAXII blockade impaired melanoma cell migration and invasion under hypoxia. This was associated with a decrease of FAK phosphorylation and metalloprotease activities. Conclusions: CAXII may be used as a target for melanoma treatment not only through its direct inhibition, but also through Hh blockade.

## 1. Introduction

Melanoma is one of the most aggressive skin cancers and represents the fifth most common cancer type in men and the sixth in women [1]. Its incidence has enhanced in recent decades [2,3,4] but unfortunately, the prognosis is quite good only for localized melanoma after surgery [5]. Indeed melanoma often metastasizes and the prognosis remains poor, along with a low free-disease survival rate [6,7]. The key role in the metastatic transition is due to increased cell motility caused by cytoskeletal changes and altered interactions with extra-cellular matrix (ECM) components [2,8].

Within this context, many new strategies have been identified in an attempt to manage malignant melanoma. This includes the targeting of the strategical FAK/paxillin pathway, whose inhibition resulted in the reduction of melanoma cell migration [9,10]. Still, with regard to ECM, the role of matrix metalloproteinase (MMPs), including MMP-9, in the rapid progression of metastatic melanoma has been also reported [11,12]. MMPs are typical components of the tumor microenvironment (TME) and the TME itself could be one of the potential targets for blocking the metastatic capability of melanoma. Nevertheless, the TME is characterized by hypoxia [13]. Indeed, the local imbalance of O_2_ consumption and supply results in the development of adaptive strategies/mechanisms that promote cell survival and progression to metastatic phenotype [14]. Among the above adaptive strategies, hypoxia triggers the expression of carbonic anhydrases (CAs) [15], a family of metalloenzymes which catalyze the reversible hydration of CO_2_ to HCO_3_^−^ and H^+^. [16]. Eight different classes have been described but only α-CAs isoforms have been characterized in humans [17]. Many roles have been attributed to CAs, from intra- and extracellular pH regulation, to homeostasis maintenance and cell survival and migration [18]. However, only human isoforms IX (CAIX) and XII (CAXII) play crucial roles in tumorigenicity and metastatic progression [19,20]. While CAIX has a limited expression in normal tissues and is upregulated in cancer, CAXII has several properties both in physiological and tumorigenic conditions, thus gaining wide interest among several researchers [21,22,23,24]. Unlike CAIX, whose role in cancer has been largely demonstrated and which has been already identified as a target in antitumor therapy [25,26,27,28,29], there are only a few reports regarding CAXII [30,31,32]. The fact that CAXII inhibition may be effective in anticancer therapy led to the synthesis of newly CAs inhibitors. In this regard, we took advantage of new available small chemical molecules which were designed starting from the pan-CAs inhibitor GV2-20 [33,34]. Novel compounds, with a different inhibitory potency against CAIX and CAXII, were identified based on an in silico drug design approach. Among them, we selected two compounds with a high specificity for the tumor-associated isoforms IX and XII: C-7 (CAIX *K*_i_ = 27.6 nM; CAXII *K*_i_ >50,000 nM) and C-10 (CAIX *K*_i_ = 16 nM; CAXII *K*_i_ = 82.1 nM). While C-7 is a selective inhibitor only for CAIX, C-10 has the most potent effect not only on CAIX but also on CAXII [32]. We have recently shown that CAXII is regulated by the Hedgehog (Hh) pathway in breast cancer [35,36]. Physiologically, the Hh pathway controls organ development during embryogenesis, whereas in adults remains quiescent, except for tissue repairing [37,38]. Of interest, aberrations of this pathway occur in tumors, being responsible for tumorigenesis and cancer maintenance [39,40]. Indeed, its role in controlling the proliferation of stem cells and tissue progenitors means the Hh pathway is increasingly the subject of cancer management studies [41,42,43,44,45].

Recent reports have shown that activation of the Hh pathway promotes metastasis in melanoma [46]. In addition, melanoma cells adapt to the acidic TME resulting in a higher tumor initiation potential [47]. Indeed, the role of CAs has already been reported in melanoma progression. However, while the involvement of CAIX has been associated with worse overall survival in patients with melanoma [48], information regarding CAXII is still scant.

For the above reasons, we decided to investigate the role of CAXII in melanoma cell migration and invasiveness. Our results, obtained by inhibiting CAXII either directly or indirectly through the Hh pathway, may contribute to the identification of novel therapeutic strategies for future management of melanoma.

## 2. Results

### 2.1. Inhibition of the Hh Pathway Affects Melanoma Cell Migration and CAXII Protein Expression

Recently, our research group discovered a correlation between the Hh pathway and CAXII on breast cancer cell migration [36]. The Hh pathway was already explored in melanoma [49,50], however its involvement in CAs modulation has not been investigated yet. Thus, we decided to analyze this possible interaction in two human melanoma cell lines, WM35 and SK-MEL-28. It should be underlined that WM35 are characterized by reduced migratory capability and aggressivity [51], while SK-MEL-28 cells are associated with intermediate aggressive phenotype, along with high cell migration and invasiveness [52]. Cells were treated with the natural smoothened (SMO) antagonist cyclopamine (cyclo) and we evaluated cell migration and CAXII protein expression. We performed experiments in both normoxic (20% O_2_ ≃ pO_2_ of 140 mmHg) and hypoxic conditions (2% O_2_ = pO_2_ of 14 mmHg) in order to mimic the TME pO_2_. As shown in Figure 1A,B, both cell lines expressed significantly higher protein levels of HIF-1α when they were exposed to hypoxia for 24 h. These results confirm that the pO_2_ employed in our experiments was adequate to induce a hypoxic state in both cell lines. We therefore investigated whether the Hh pathway was involved in melanoma cell migration, either under normoxia or hypoxia. To this end, we performed a wound healing assay, where confluent monolayers of WM35 and SK-MEL-28 were scratched after an overnight treatment with cyclo and cell motility was monitored for 24 h incubation either under normoxic or hypoxic conditions. Of interest, inhibition of the Hh pathway did not significantly affect the WM35 cell migration rate (Figure 1B,C), either in normoxia or hypoxia. However, Hh inhibition resulted in a significant reduction of SK-MEL-28 cell migration in a normoxic environment. Indeed, we observed a five-fold inhibition of cell migration after 24 h treatment when compared with the untreated cells (Figure 1B). Interestingly, the impairment of cell migration was significant also under hypoxia (Figure 1C), indicating that Hh inhibition was also effective at a pO_2_ which resembles the TME. To investigate whether the Hh pathway may be involved in modulating the expression of CAXII in melanoma cells, we then analyzed the protein levels of CAXII in both cell lines exposed either under normoxia or hypoxia, after 24 h treatment in the presence or not of cyclo. Figure 1D clearly shows that WM35 cells expressed low levels of CAXII. Thereafter, we did not observe any significant differences upon cyclo treatment in both experimental conditions. In contrast, CAXII expression was significantly increased in hypoxic SK-MEL-28, and according with the cell migration results, cyclo treatment significantly reduced the level of CAXII (Figure 1D). The latter results indicate that in SK-MEL-28 cell line the expression of CAXII is regulated by the Hh pathway. The same regulatory effect may be present also in WM35, but the low protein levels do not allow the detection of a significant reduction in protein expression.

### 2.2. Effects of CAs Inhibitors on Melanoma Cell Migration

Since inhibition of the Hh pathway by cyclo impaired both cell migration and CAXII expression in SK-MEL-28, we next evaluated whether a direct inhibition of CAXII could affect melanoma cell migration as well. To this end, we used not only WM35 and SK-MEL28 but also a third cell line, A375, which is characterized by a highly aggressive phenotype.

First of all, we measured CAXII protein expression in all the cell lines, under normoxia or hypoxia, to ascertain a different expression within the two microenvironments. Figure 2A shows that only SK-MEL-28 expressed high level of CAXII under normoxic condition. As expected, under hypoxia, CAXII expression in WM35 remained low; however, its expression was still high in SK-MEL-28 and, of interest, it was significantly upregulated in A375. Thus, the different CAXII expression, especially in hypoxic conditions, seemed to be related to the different aggressiveness of the cell lines. Thereafter, we used two new small molecules, C-7 and C-10, along with the pan-CAs inhibitors GV2-20 and acetazolamide (AAZ), which were employed as reference compounds. As described above, C-7 is a selective CAIX inhibitor, while C-10 inhibits both CAIX and CAXII at nanomolar concentrations [33]. First of all, we tested the effects of these new compounds on WM35, SK-MEL-28, and A375 viability up to 72 h incubation. As shown in Figure 2B, CAs inhibitors did not significantly affect cell viability in all cell lines at 10 nM concentration, which was the same concentration employed in the following assays. This was evident when the cell lines were exposed either to normoxia or hypoxia. These results were important to rule out the possibility that the migration and invasion assays might be related to cell viability variations.

Thus, we evaluated the effect of CAs inhibitors on cell migration by performing a scratch assay. Similarly to the Hh inhibition results, pretreatment with CAs inhibitors did not significantly affect WM35 migration either under normoxia or hypoxia (Figure 2C,D). However, we observed a significant reduction in SK-MEL-28 and A375 cell migration, after pretreatment with compound C-10. Again these effects were similar under either normoxic or hypoxic conditions (Figure 2C,D). These results suggest that the inhibition of CAIX alone is not sufficient to regulate cell migration. In contrast, CAXII appears to be a key regulator in the migration of the intermediate and high aggressive melanoma cell lines SK-MEL-28 and A375. Indeed, C-10 is characterized as one of the more specific inhibitors of CAXII, when compared not only to C-7 but also to the pan-inhibitors GV2-20 and AAZ.

### 2.3. SK-MEL-28 and A375 Invasion Is Reduced by Direct CAXII Inhibition under Hypoxia

Based on the above results, showing that CAs inhibitors significantly affected only SK-MEL-28 and A375 migration, we next focused our experiments on these two, more aggressive cell lines, which were exposed to a hypoxic condition that resembled the TME. It should be underlined that previous results have shown that hypoxia significantly affects the invasiveness of melanoma cells [53,54,55,56,57].

At first, we evaluated the effect of CAs inhibition on hypoxic SK-MEL-28 and A375 invasiveness. Figure 3A clearly shows that C-10, unlike the pan-inhibitor AAZ, was able to reduce SK-MEL-28 invasion in a significant manner. A375 invasion was also significantly impaired by both C-7 and C-10 inhibitors but the last resulted in the most potent effect. These results indicate that the inhibition of CAXII plays a crucial role in regulating the invasiveness of both intermediate and highly aggressive melanoma cell lines. Furthermore, we analyzed the effects of C-7 and C-10 on the phosphorylated focal adhesion kinase (phFAK), as its role in the promotion of the aggressive melanoma phenotype has been previously established in several reports [58]. In line with the migration and invasion results, inhibition by C-10 resulted in a significant decrease of phFAK levels both in SK-MEL-28 and A375 (Figure 3B). In addition, C-10 treatment resulted also in a significant decrease of MMP-9 activity, as shown by the zymographic analyses in Figure 3C. The relevance of MMP-9 in melanoma malignancy and progression has been already documented [11]. However, such an inhibition was exerted also by C-7, indicating that CAIX and CAXII were similarly involved in the regulation of MMP-9. Finally, to further confirm the inhibitory effect of C-7 and C-10 on CAIX and CAXII, we analyzed their protein expression in hypoxic SK-MEL28 by fluorescence microscopy. As shown in Figure 3D, C-7 and C-10 were able to reduce CAIX expression. However, the expression of CAXII was significantly reduced only by C-10. The fluorescence data supported our results which demonstrated that C10 was able to reduce the expression of CAXII, which could be crucial in the regulation of melanoma cell migration and invasion.

## 3. Discussion

CAXII is a CAs isoform strictly related to cancer and hypoxic responses. In fact, when oxygen is lacking, CAXII expression is modified, as tumor cells might still proliferate and survive [26]. CAXII has been recently proposed as a possible target for anticancer therapy [31] and novel and specific inhibitors of this enzyme, along with CAIX, have been recently identified [33]. Furthermore, we have recently reported that CAXII expression is regulated by the Hh pathway in breast cancer, indicating that this enzyme plays a potential role in breast cancer cell migration and invasiveness [36]. We here report, for the first time, that the Hh pathway is involved in the modulation of CAXII also in melanoma cells, in particular, in a moderate aggressive melanoma cell line and, more interestingly, in hypoxic conditions.

Previous reports clearly demonstrated that hypoxia was critical for tumor progression in melanoma [57]. In the present manuscript, we show that the more aggressive melanoma cell lines, SK-MEL-28 and A375, expressed higher protein levels of CAXII under hypoxia. This is in agreement with previous reports showing that CAXII expression is increased in several aggressive tumors along with a poor prognosis [27]. Furthermore, hypoxia has been reported to trigger the Hh pathway in colorectal cancer [59], suggesting that its inhibition may be relevant for tumor progression. In the present manuscript, we show that Hh inhibition, under hypoxia, resulted in a downregulation of CAXII. Moreover, we here report that CAXII is overexpressed in SK-MEL-28 and A375 in comparison to WM35 and that its expression correlates with higher migration capability. Thus, CAXII appears to be crucial for cell migration. It should be underlined that cell migration is associated with epithelial–mesenchymal transition (EMT) in several neoplasias, including the metastatic progression in melanoma [60,61,62,63]. The involvement of the Hh pathway was already explored in melanoma and its blockade reduced cell viability and migration [49,50]. The fact that, in the present manuscript, Hh inhibition resulted in CAXII downregulation in melanoma cells, and in particular in the more aggressive cell line, opens a new scenario for a possible therapeutic targeting. In fact, C-10, which is active at nanomolar concentrations on both CAXII and CAIX, was able to reduce cell migration while the inhibition of only CAIX by C-7 was not sufficient. It should be pointed out that in our experimental protocol C-10 was even more effective than the pan-inhibitors GV2-20 and AAZ. We and others have previously reported that CAXII promotes not only the migration but also invasive capability of breast cancer cells [36,64]. Indeed, the decreased invasiveness and migration ability of CAXII-knockdown cells were restored by an overexpression of CAXII [64]. Here, we demonstrated that C-10 was particularly effective in reducing the invasion ability of SK-MEL-28 and A375 under a hypoxic condition which mimics the TME. Furthermore, we here report that C-10 reduced the phosphorylation of FAK as well as the activity of MMP-9. Since both FAK and MMP-9 have been previously related to the invasive properties of melanoma cells, our results further support the anti-invasive properties of CAXII inhibition.

## 4. Materials and Methods

### 4.1. Cell Cultures

WM35 and SK-MEL-28 were kindly donated by Dr. Raffaella Giavazzi (Department of Oncology, Istituto di Ricerche Farmacologiche Mario Negri IRCCS, Milan, Italy) and Dr. Francesca Chiarini (CNR Institute of Molecular Genetics “Luigi Luca Cavalli-Sforza”, Bologna, Italy.) respectively. A375 were purchased from American Type Culture Collection (ATCC, Manassas, VA, USA). WM35 and SK-MEL-28 were maintained in RPMI 1640 (Euroclone, Devon, UK) and A375 were cultured in Dulbecco’s modified Eagle’s medium (DMEM, Euroclone, Devon, UK). All media were supplemented with antibiotics, L-glutamine 2 mM and 10% fetal bovine serum (FBS) (Euroclone, Devon, UK), at 37 °C in humidified 5% CO_2_.

The hypoxic treatment was performed using a workstation In Vivo 400 (Ruskinn, Pencoed, UK) providing a customized and stable humidified environment through electronic control of 2% O_2_ and 5% CO_2_.

### 4.2. Chemical Compounds

C-7 and C-10 are new small molecules targeting CAIX or both CAIX and CAXII respectively, which were kindly provided by Department of Biotechnology, Chemistry and Pharmacy (University of Siena, Siena, Italy) [33].

We used pan-CAs inhibitor GV2-20 (a benzoic acid derivative previously identified as CAs inhibitor by a computational-driven target fishing approach [34] (Molport, Riga, Latvia)) and AAZ (Sigma-Aldrich, St.Louis, MO, USA) as reference compounds.

All the inhibitors were resuspended in dimethyl sulfoxide (DMSO) and diluted in water and used at a final concentration of 10 nM.

Cyclo (AlphaAesar, Haverhill, MA, USA) was resuspended in DMSO and used at the final concentration of 20 µM.

### 4.3. Cell Viability

Cells were seeded in 96-well plates at a density of 2 × 10^4^ cells per well, treated with CAs inhibitors and incubated under normoxia or hypoxia. After 72 h, they were counted using fluorescein diacetate (1 mg/mL, diluted 1:200). The viability index was calculated as the percentage of the ratio between treated groups and control. All the experiments were run in triplicate and repeated at least three times.

### 4.4. Western Blot

Cells were seeded in 60 mm petri dishes at a density of 5 × 10^5^ cells per dish and, following specific treatments, were incubated for 24 h in normoxic or hypoxic condition. Thereafter, cells were washed with cold PBS, were lysed with Laemmli Buffer containing protease inhibitors (1 mM PMSF, 1 mg/mL aprotinin, 1 mg/mL leupeptin) and frozen at −20 °C. Samples were sonicated for three times and the quantification of total protein concentration was determined using Micro BCA Protein Assay Reagent kit (Thermo Fisher Scientific, Cleveland, OH, USA). Sample measures of 40 µg per lane were resolved on 10% acrylamide gel, transferred to nitrocellulose membrane, and incubated for 1 h with 5% nonfat powdered milk. We used HIF-1α (BD Biosciences, San Jose, CA, USA), phFAK, FAK (Cell Signaling, Denver, CO, USA), CAXII (Santa Cruz, CA, USA), and β-Actin (Sigma-Aldrich, St. Louis, MO, USA). The specific secondary antibodies, anti-rabbit IgG-HRP and anti-mouse IgG-HRP (Cell Signaling, Denver, CO, USA), were incubated for 1 h. Chemiluminescence of immunoreactive bands was detected with ChemiDoc™ MP System (Bio-Rad, Hercules, CA, USA) and quantified with Image Lab software. The original Western blot figures can be found in the supplementary file.

### 4.5. Scratch Wound Healing Assay

Cells were seeded in 24-well plates at a density of 1 × 10^5^ cells per well, pretreated with CAs inhibitors overnight and the following day the confluent monolayer was scratched with a 200 µL tip. Washing twice with PBS and refreshing medium with 10% FBS was followed. Each well was treated again with the respective inhibitor and incubated either under normoxic or hypoxic microenvironments. After 0 h and 24 h snapshots of the scratch were taken with the microscope (10×) (Olympus IX81, Tokyo, Japan) with cell F© software. Migration was calculated as $(1-\frac{Ax}{A0})\%$, where *A*0 and *Ax* represented the empty scratch area at 0 h and 24 h, respectively.

### 4.6. Boyden Chamber

Chemoinvasion was analyzed employing a modified Boyden 48-well micro chemotaxis chamber (Neuro Probe, Gaithersburg, MD, USA) with 8 μm pore size polycarbonate polyvinylpyrrolidone-free nucleopore filters, as previously described [65,66]. Cells were pretreated overnight, the following day were resuspended in RPMI 0,1% bovine serum albumin (BSA) and 50 µL of an amount of 2.7 × 10^5^ cell/mL were loaded in triplicate on each upper well, which was previously coated with 0.5 mg/mL of Matrigel (Corning, Life Science, Corning, Tewksbury, MA, USA). NIH3T3 supernatant was added in the lower chamber and used as chemoattractant. After 24 h incubation under hypoxic condition, cells on the lower surface were stained using Diff Quick (Merz-Dade, Düdingen, Switzerland) and photographed by an OLYMPUS IX81 Research Microscope with a 10× magnification. Invaded cells in four high-power fields were counted and data were expressed as number of invaded cells per field.

### 4.7. Zymography

Cells were incubated with C-7 and C-10 inhibitors for 24 h under hypoxia and then media were collected and centrifuged at 300× *g* for 10 min to remove cells debris. Total protein amount was analyzed using Micro BCA Protein Assay Reagent kit (Thermo Fisher Scientific, Cleveland, OH, USA) and 15 µg of each appropriately diluted media was loaded into 0.1% gelatin-8% acrylamide gels. After electrophoresis, gelatin gels were washed with 2.5% Triton x-100 for 30 min in agitation and an overnight incubation at 37 °C in agitation with developing buffer was followed. Then, gels were stained with Blue Comassie R-250 (Sigma Aldrich, St.Louis, MO, USA) and, after destaining, images were taken with ChemiDoc™ MP System (Bio-Rad, Hercules, CA, USA) and quantified with ImageJ software.

### 4.8. Immunofluorescence Staining

SK-MEL-28 were plated on 11 mm diameter glass coverslips in a 24-well plate at 10^5^ cells per well and treated with C-7 and C-10 inhibitors for 24 h before fixation. Cells were then fixed in cold methanol for 10 min at −20 °C, were then washed three times for 10 min with PBS and incubated in blocking buffer (1% BSA in PBS) for 30 min at room temperature. An overnight incubation in CAIX (Cell Signaling, Denver, CO, USA, diluted 1:10 in PBS/BSA) or CAXII primary antibody (Santa Cruz, CA, USA; diluted 1:50 in PBS/BSA) at 4 °C in humid chambers was followed. Coverslips were then washed for 10 min in PBS/BSA and incubated with secondary antibody (Alexa Fluor-555-conjugated anti-rabbit-IgG and Alexa Fluor-488-conjugated anti-mouse-IgG, 1:800; Invitrogen, Carlsbad, CA, USA) for 1 h at room temperature. After a final washing in PBS, the coverslips were mounted on slides with 90% glycerol in PBS. DNA was visualized with incubation of 3–4 min in Hoechst (Sigma Aldrich, St.Louis, MO, USA).

Images were taken by using an Axio Imager Z1 (Carl Zeiss, Jena, Germany) microscope equipped with an AxioCam HR cooled charge-coupled camera (Carl Zeiss, Jena, Germany). Grayscale digital images were collected separately and pseudocolored and merged using Adobe Photoshop 7.0 software (Adobe Incorporated, San Jose, CA, USA). Fluorescence intensity was measured with ImageJ software.

### 4.9. Statistical Analyses

Data are presented as means ± standard error of the means of at least three independent experiments performed in duplicates or triplicates. Statistical analyses were conducted using unpaired *t*-test and ANOVA test with GraphPad Prism 7 software. Values of *p* ≤ 0.05 and *p* ≤ 0.01 were conventionally considered statistically significant.

## 5. Conclusions

The identification of new small molecules with specific inhibitory activities on different CAs, especially CAXII, opens a new scenario in the therapeutic approach in melanoma patients. Previous reports indicate the relevance of hypoxia and CAs in the TME and metastatic progression. Overall, our results suggest that CAXII inhibition under hypoxia leads to a reduction of both cell migration and invasion in aggressive melanoma cell lines. This, along with our data regarding Hh inhibition, may contribute to possible alternative treatment by acting either directly or indirectly on CAXII to reduce the metastatic capability of melanoma.

## Figures and Tables

**Figure 1 cancers-12-03018-f001:**
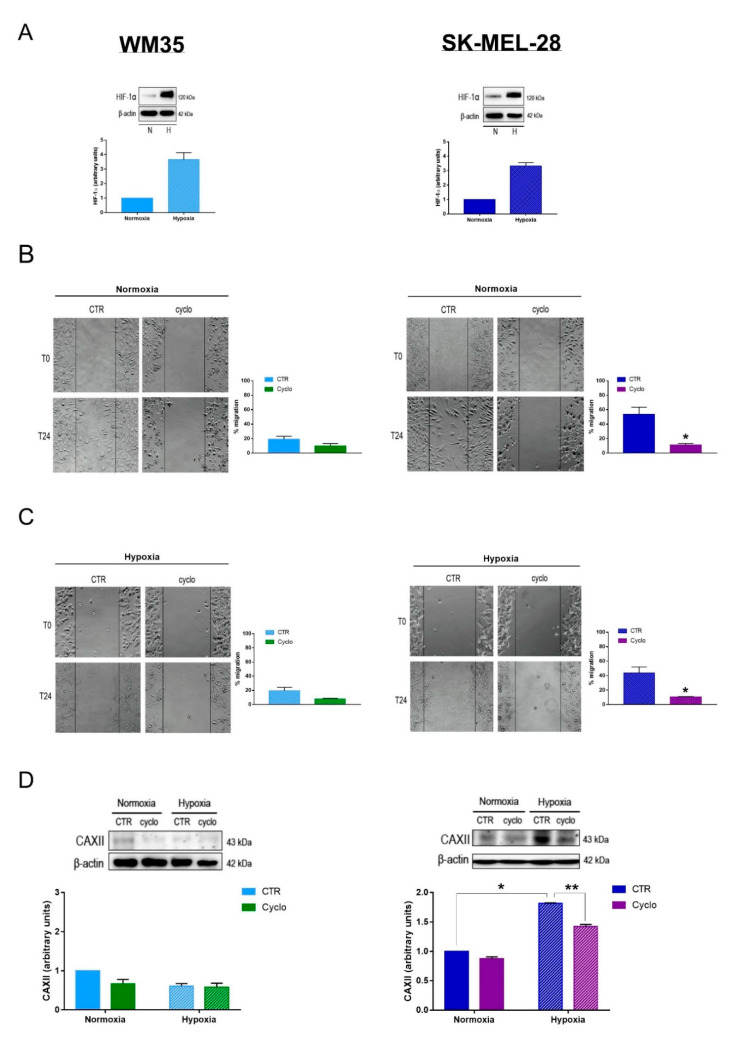
Inhibition of the Hedgehog pathway affects melanoma cell migration and CAXII protein expression. (**A**) HIF-1α protein expression in WM35 and SK-MEL-28 after 24 h exposure to normoxia or hypoxia. Blots are representations of the means of three independent experiments and β-actin was used as loading control. (**B**) Representative image of WM35 and SK-MEL-28 scratch assay after 24 h treatment with 20 µM cyclopamine (cyclo) under normoxia. Phase contrast microscopy images were taken with Olympus IX81 at a 10× magnification. Quantifications represent the means and standard errors of the mean of three independent experiments. (**C**) Representative image of WM35 and SK-MEL-28 scratch assay after 24 h treatment with 20 µM cyclo under hypoxia. Phase contrast microscopy images were taken with Olympus IX81 at a 10× magnification. Quantifications represent the means and standard errors of the mean of three independent experiments. (**D**) CAXII protein expression in WM35 and SK-MEL-28 after 24 h treatment with 20 µM cyclo exposed to normoxia or hypoxia. Blots are representation of three independent experiments and β-actin was used as loading control. * *p* ≤ 0.05, ** *p* ≤ 0.01 indicate statistically significant differences. The uncropped blots of Figure 1A,D are shown in Appendix A.

**Figure 2 cancers-12-03018-f002:**
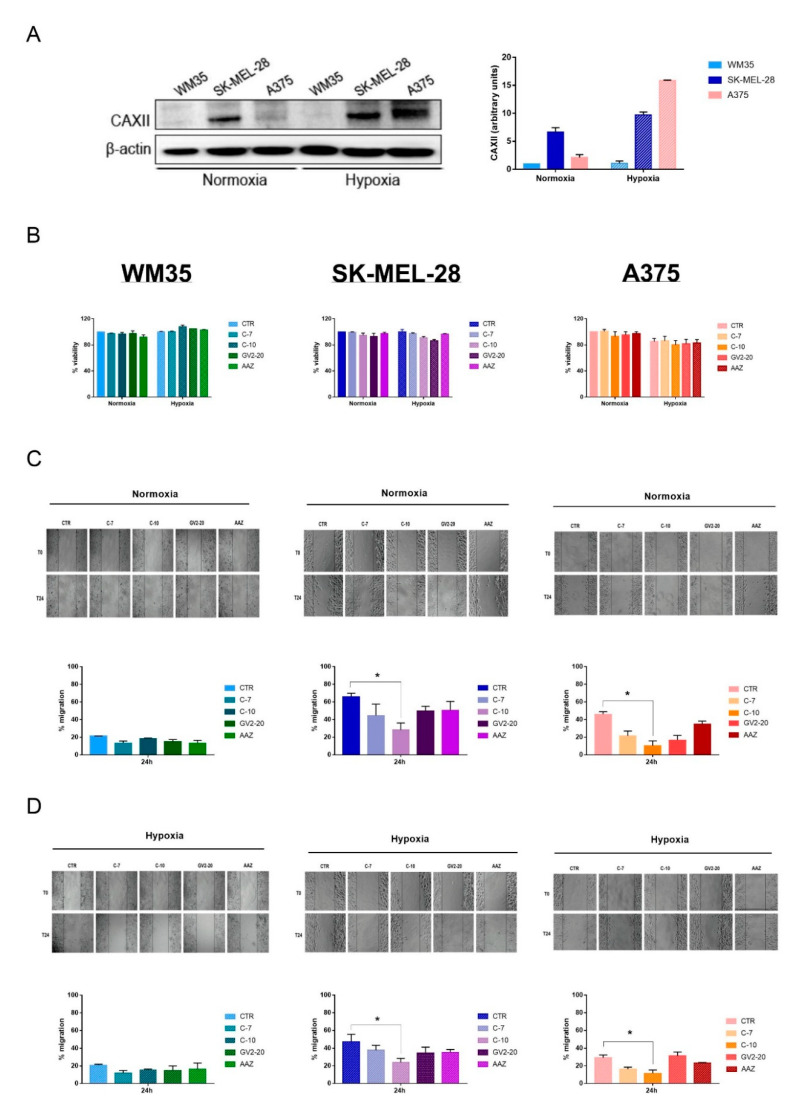
Effects of CAs inhibitors on cell migration. (**A**) CAXII protein expression in WM35, SK-MEL-28, and A375 under normoxia and hypoxia. Blot is a representation of the means and standard errors of the mean of three independent experiments. β-actin was used as loading control. (**B**) WM35, SK-MEL-28, and A375 viability assays after 72 h treatment with 10 nM of each compound and exposed to normoxia or hypoxia. Data are reported as the percentage of the ratio between the treated groups and controls. Graphs represent the means and standard errors of the mean of three independent experiments performed in triplicates. (**C**) WM35, SK-MEL-28, and A375 scratch assays after an overnight pretreatment and 24 h treatment with 10 nM of each compound and exposed to normoxia. Phase contrast microscopy images were taken with Olympus IX81 at a 10× magnification. Data are represented as the percentage of migration rate between treated groups and controls. Graphs represent the means and standard errors of the mean of three independent experiments performed in duplicates. (**D**) WM35, SK-MEL-28 and A375 scratch assays after an overnight pretreatment and 24 h treatment with 10 nM of each compound and exposed to hypoxia. Phase contrast microscopy images were taken with Olympus IX81 at a 10× magnification. Data are represented as the percentage of migration rate between treated groups and controls. Graphs represent the means and standard errors of the mean of three independent experiments performed in duplicates. * indicates statistically significant differences (*p* ≤ 0.05). The uncropped blots of Figure 2A is shown in Appendix A.

**Figure 3 cancers-12-03018-f003:**
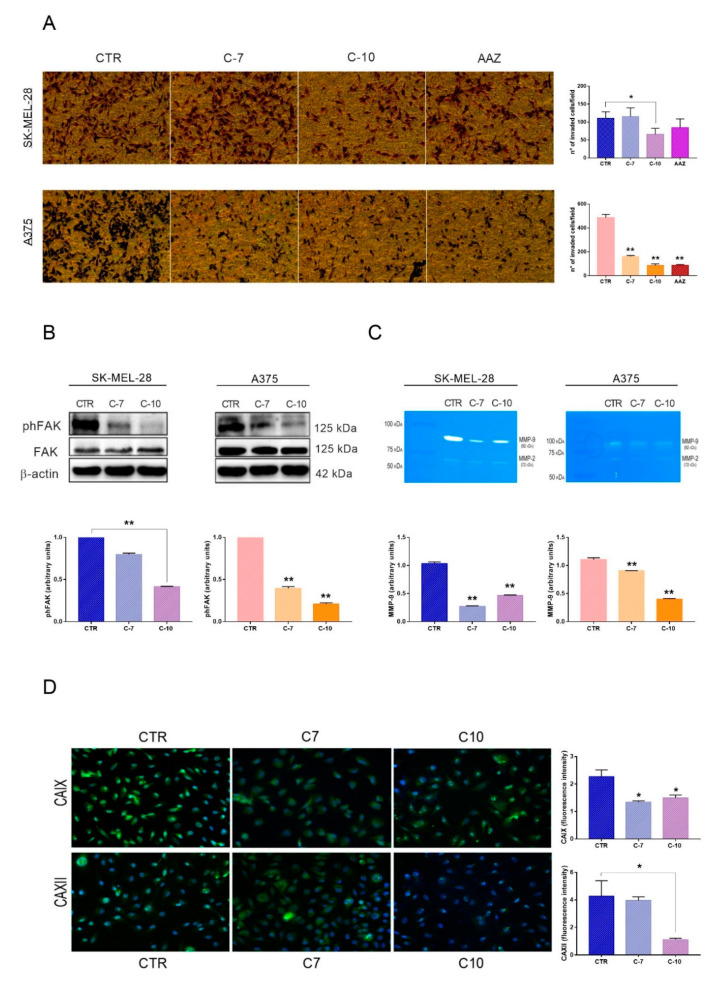
SK-MEL-28 and A375 invasion is reduced by direct CAXII inhibition under hypoxia. (**A**) Modified Boyden chamber invasion assays after pretreatment and 24 h treatment under hypoxia (10× magnification). Data are presented as the means and standard errors of the mean of three independent experiments performed in triplicates. Acetazolamide (AAZ) was used as control. (**B**) Phosphorylated focal adhesion kinase (phFAK) protein expressions after 24 h treatment with C-7 and C-10 under hypoxia as determined by western blot analyses. Data are presented as the means and standard errors of the mean of three independent experiments. Β-actin was used as loading control. (**C**) Matrix metalloproteinase 9 (MMP-9) enzymatic activity after 24 h treatment upon hypoxia as determined by zymography. Data are presented as the means and standard errors of the mean of three independent experiments. (**D**) CAXII protein expression in SK-MEL-28 after 24 h treatment with C-7 and C-10 under hypoxia as determined by immunofluorescence microscopy analyses. (25× objective 50× magnification). Data are presented as the means and standard errors of the mean of three independent experiments. * *p* ≤ 0.05, ** *p* ≤ 0.01 indicate statistically significant differences. The uncropped blots of Figure 3B is shown in Appendix A.

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
