# Peer review of "Inhibition of Melanoma Cell Migration and Invasion Targeting the Hypoxic Tumor Associated CAXII"

_cancers, 2020, doi:10.3390/cancers12103018_

Round 1

Reviewer 1 Report

Authors found that carbonic anhydrases IX which is hypoxia-related protein, inhibits melanoma cell migration and invasion under hypoxia condition through Hedgehog pathway or directly. Experiment and methods are good and logical. But, there are a minor point that could be addressed as follows. What is the reason of different results between cell lines? What if you wait for longer time to see the slow late reaction? Do you think hypoxia or CAXII may effect on only rapid growing aggressive melanomas, except slow growing melanomas? Add the explains and longer effect results.

Author Response

First of all, we thank the reviewer for the positive comments. Our hypothesis is that we obtained different results for different cell lines because of the different migration capacities and different CAXII expression, as reported in fig 2 of the revised manuscript.

We thank the reviewer for having underlined the prolonged treatment issue: A 48h treatment did not result in a prolonged inhibition of cell migration (data not shown). In our protocol the compounds were added only at the beginning of the experiments. We did not observed a significant effect after more than 24 h for the fast turnover rate of CA protein and the plausible degradation-inactivation of the compounds.

With regard to the last point, we believed that only rapid growing aggressive cells are affected by the inhibition of CA XII. To confirm our hypothesis, we added results where inhibition of CAXII was able to reduce the invasion capability of another highly aggressive melanoma cell line (A375) beside the SK-MEL-28 (see figg 2-3 in the revised manuscript)

Reviewer 2 Report

In their work the authors nicely show the impact of the carbonic anhydrases IX and especially XII on the migration and invasion of melanoma cells. Hh inhibition reduces melanoma migration under both normoxic and hypoxic 27 conditions. Similarly, CAXII blockade impairs melanoma migration and invasion upon hypoxia.

The authors conclude that CAXII may be used as a target for melanoma treatment not only through its direct inhibition, but also through Hh blockade.

In the beginning they used two cell lines from which only the SKMel28 had an adequate amount of CAXII and was followed further on. I suggest that the authors must proof their hypothesis at least in one other cell line with an adequate amount of CAXII. From the broad institute data it is known that there are several melanoma cell lines for which this applies (for example UACC62, WM266-4 or Malme-3M), or which show an increased mRNA expression of CAXII. With one of these cell lines the authors should proof their theory and show that C10 is able to inhibit the migratory potential of the cells under normoxic and hypoxic conditions.

Author Response

We would like to thank the reviewer for the comment. Accordingly, in the revised manuscript we added results obtained by another highly aggressive melanoma cell line, A375. First we showed that this cell line expressed adequate amount of CAXII (see fig.2 of the revised manuscript). Thereafter, we reported that, similarly to SK-MEL-28 cell line, CAXII blockade impairs A375 cell migration and invasion upon hypoxia (see figg. 2-3 of the revised manuscript).

Reviewer 3 Report

In their article, Giuntini et al. investigate an important question addressing the contribution of hypoxia to cancer progression. They focus in their research on the important role of carbonic anhydrases IX (CAIX) and XII (CAXII) and were able to demonstrate that the mobility of melanoma cells is impaired by direct targeting CAXII or through the inhibition of the aberrant Hedgehog pathway.

The authors used two different melanoma cell lines showing different phenotypes in terms of migratory capacity and invasiveness, WM35 and SK-MEL28. The CAXII protein levels did not change in hypoxic WM35 cells, whereas hypoxic SK-MEL 28 cells showed a significant increase in CAXII. Only hypoxic SK-MEL cells showed a decreased migration and invasion capacity when exposed to CA inhibitors. These data provide experimental preclinical evidence, that direct targeting of CAXII in a specific subgroup of melanoma patient might be beneficial.

The manuscript is very well written and concise, the experiments are well designed.

Author Response

We thank the reviewer for the very positive comments.

Round 2

Reviewer 2 Report

The authors addressed all critical points.